# MIXED PRECISION TRAINING

**Sharan Narang**[*], **Gregory Diamos, Erich Elsen**[†]
Baidu Research
{sharan, gdiamos}@baidu.com

**Paulius Micikevicius**[*], **Jonah Alben, David Garcia, Boris Ginsburg, Michael Houston,
Oleksii Kuchaiev, Ganesh Venkatesh, Hao Wu**
NVIDIA
{pauliusm, alben, dagarcia, bginsburg, mhouston,
 okuchaiev, gavenkatesh, skyw}@nvidia.com

## ABSTRACT

Increasing the size of a neural network typically improves accuracy but also increases the memory and compute requirements for training the model. We introduce methodology for training deep neural networks using half-precision floating point numbers, without losing model accuracy or having to modify hyperparameters. This nearly halves memory requirements and, on recent GPUs, speeds up arithmetic. Weights, activations, and gradients are stored in IEEE half-precision format. Since this format has a narrower range than single-precision we propose three techniques for preventing the loss of critical information. Firstly, we recommend maintaining a single-precision copy of weights that accumulates the gradients after each optimizer step (this copy is rounded to half-precision for the forward- and back-propagation). Secondly, we propose loss-scaling to preserve gradient values with small magnitudes. Thirdly, we use half-precision arithmetic that accumulates into single-precision outputs, which are converted to half-precision before storing to memory. We demonstrate that the proposed methodology works across a wide variety of tasks and modern large scale (exceeding 100 million parameters) model architectures, trained on large datasets.

## 1 INTRODUCTION

Deep Learning has enabled progress in many different applications, ranging from image recognition (He et al., 2016a) to language modeling (Jozefowicz et al., 2016) to machine translation (Wu et al., 2016) and speech recognition (Amodei et al., 2016). Two trends have been critical to these results - increasingly large training data sets and increasingly complex models. For example, the neural network used in Hannun et al. (2014) had 11 million parameters which grew to approximately 67 million for bidirectional RNNs and further to 116 million for the latest forward only Gated Recurrent Unit (GRU) models in Amodei et al. (2016).

Larger models usually require more compute and memory resources to train. These requirements can be lowered by using reduced precision representation and arithmetic. Performance (speed) of any program, including neural network training and inference, is limited by one of three factors: arithmetic bandwidth, memory bandwidth, or latency. Reduced precision addresses two of these limiters. Memory bandwidth pressure is lowered by using fewer bits to to store the same number of values. Arithmetic time can also be lowered on processors that offer higher throughput for reduced precision math. For example, half-precision math throughput in recent GPUs is $2\times$ to $8\times$ higher than for single-precision. In addition to speed improvements, reduced precision formats also reduce the amount of memory required for training.

Modern deep learning training systems use single-precision (FP32) format. In this paper, we address the training with reduced precision while maintaining model accuracy. Specifically, we train vari-

---

[*]Equal contribution
[†]Now at Google Brain eriche@google.com

ous neural networks using IEEE half-precision format (FP16). Since FP16 format has a narrower dynamic range than FP32, we introduce three techniques to prevent model accuracy loss: maintaining a master copy of weights in FP32, loss-scaling that minimizes gradient values becoming zeros, and FP16 arithmetic with accumulation in FP32. Using these techniques we demonstrate that a wide variety of network architectures and applications can be trained to match the accuracy FP32 training. Experimental results include convolutional and recurrent network architectures, trained for classification, regression, and generative tasks. Applications include image classification, image generation, object detection, language modeling, machine translation, and speech recognition. The proposed methodology requires no changes to models or training hyper-parameters.

## 2   RELATED WORK

There have been a number of publications on training Convolutional Neural Networks (CNNs) with reduced precision. Courbariaux et al. (2015) proposed training with binary weights, all other tensors and arithmetic were in full precision. Hubara et al. (2016a) extended that work to also binarize the activations, but gradients were stored and computed in single precision. Hubara et al. (2016b) considered quantization of weights and activations to 2, 4 and 6 bits, gradients were real numbers. Rastegari et al. (2016) binarize all tensors, including the gradients. However, all of these approaches lead to non-trivial loss of accuracy when larger CNN models were trained for ILSVRC classification task (Russakovsky et al., 2015). Zhou et al. (2016) quantize weights, activations, and gradients to different bit counts to further improve result accuracy. This still incurs some accuracy loss and requires a search over bit width configurations per network, which can be impractical for larger models. Mishra et al. improve on the top-1 accuracy achieved by prior weight and activation quantizations by doubling or tripling the width of layers in popular CNNs. However, the gradients are still computed and stored in single precision, while quantized model accuracy is lower than that of the widened baseline. Gupta et al. (2015) demonstrate that 16 bit fixed point representation can be used to train CNNs on MNIST and CIFAR-10 datasets without accuracy loss. It is not clear how this approach would work on the larger CNNs trained on large datasets or whether it would work for Recurrent Neural Networks (RNNs).

There have also been several proposals to quantize RNN training. He et al. (2016c) train quantized variants of the GRU (Cho et al., 2014) and Long Short Term Memory (LSTM) (Hochreiter and Schmidhuber, 1997) cells to use fewer bits for weights and activations, albeit with a small loss in accuracy. It is not clear whether their results hold for larger networks needed for larger datasets Hubara et al. (2016b) propose another approach to quantize RNNs without altering their structure. Another approach to quantize RNNs is proposed in Ott et al. (2016). They evaluate binary, ternary and exponential quantization for weights in various different RNN models trained for language modelling and speech recognition. All of these approaches leave the gradients unmodified in single-precision and therefore the computation cost during back propagation is unchanged.

The techniques proposed in this paper are different from the above approaches in three aspects. First, all tensors and arithmetic for forward and backward passes use reduced precision, FP16 in our case. Second, no hyper-parameters (such as layer width) are adjusted. Lastly, models trained with these techniques do not incur accuracy loss when compared to single-precision baselines. We demonstrate that this technique works across a variety of applications using state-of-the-art models trained on large scale datasets.

## 3   IMPLEMENTATION

We introduce the key techniques for training with FP16 while still matching the model accuracy of FP32 training session: single-precision master weights and updates, loss-scaling, and accumulating FP16 products into FP32. Results of training with these techniques are presented in Section 4.

### 3.1   FP32 MASTER COPY OF WEIGHTS

In mixed precision training, weights, activations and gradients are stored as FP16. In order to match the accuracy of the FP32 networks, an FP32 master copy of weights is maintained and updated with the weight gradient during the optimizer step. In each iteration an FP16 copy of the master weights is

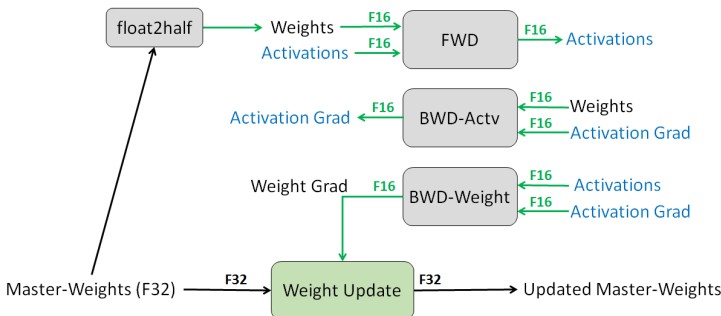

Figure 1: Mixed precision training iteration for a layer.

used in the forward and backward pass, halving the storage and bandwidth needed by FP32 training. Figure 1 illustrates this mixed precision training process.

While the need for FP32 master weights is not universal, there are two possible reasons why a number of networks require it. One explanation is that updates (weight gradients multiplied by the learning rate) become too small to be represented in FP16 - any value whose magnitude is smaller than $2^{-24}$ becomes zero in FP16. We can see in Figure 2b that approximately 5% of weight gradient values have exponents smaller than $-24$. These small valued gradients would become zero in the optimizer when multiplied with the learning rate and adversely affect the model accuracy. Using a single-precision copy for the updates allows us to overcome this problem and recover the accuracy.

Another explanation is that the ratio of the weight value to the weight update is very large. In this case, even though the weight update is representable in FP16, it could still become zero when addition operation right-shifts it to align the binary point with the weight. This can happen when the magnitude of a normalized weight value is at least 2048 times larger that of the weight update. Since FP16 has 10 bits of mantissa, the implicit bit must be right-shifted by 11 or more positions to potentially create a zero (in some cases rounding can recover the value). In cases where the ratio is larger than 2048, the implicit bit would be right-shifted by 12 or more positions. This will cause the weight update to become a zero which cannot be recovered. An even larger ratio will result in this effect for de-normalized numbers. Again, this effect can be counteracted by computing the update in FP32.

To illustrate the need for an FP32 master copy of weights, we use the Mandarin speech model (described in more detail in Section 4.3) trained on a dataset comprising of approximately 800 hours of speech data for 20 epochs. As shown in 2a, we match FP32 training results when updating an FP32 master copy of weights after FP16 forward and backward passes, while updating FP16 weights results in 80% relative accuracy loss.

Even though maintaining an additional copy of weights increases the memory requirements for the weights by 50% compared with single precision training, impact on overall memory usage is much smaller. For training memory consumption is dominated by activations, due to larger batch sizes and activations of each layer being saved for reuse in the back-propagation pass. Since activations are also stored in half-precision format, the overall memory consumption for training deep neural networks is roughly halved.

## 3.2 LOSS SCALING

FP16 exponent bias centers the range of normalized value exponents to $[-14, 15]$ while gradient values in practice tend to be dominated by small magnitudes (negative exponents). For example, consider Figure 3 showing the histogram of activation gradient values, collected across all layers during FP32 training of Multibox SSD detector network (Liu et al., 2015a). Note that much of the FP16 representable range was left unused, while many values were below the minimum representable range and became zeros. Scaling up the gradients will shift them to occupy more of the representable range and preserve values that are otherwise lost to zeros. This particular network diverges when gradients are not scaled, but scaling them by a factor of 8 (increasing the exponents by 3) is sufficient to match the accuracy achieved with FP32 training. This suggests that activation

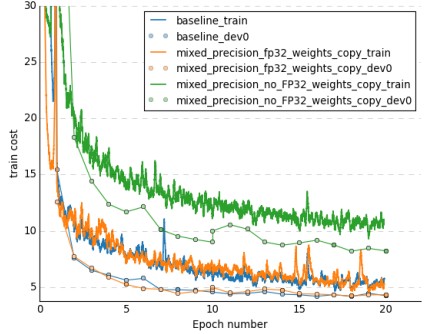

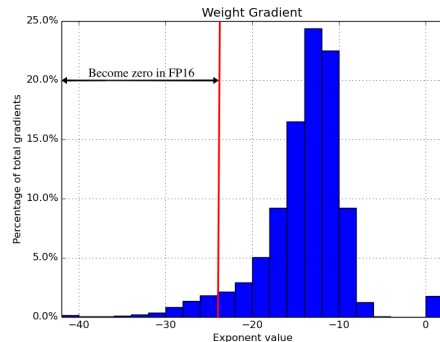

(a) Training and validation (dev0) curves for Mandarin speech recognition model

(b) Gradient histogram for Mandarin training run

Figure 2: Figure 2a shows the results of three experiemnts; baseline (FP32), pseudo FP16 with FP32 master copy, pseudo FP16 without FP32 master copy. Figure 2b shows the histogram for the exponents of weight gradients for Mandarin speech recognition training with FP32 weights. The gradients are sampled every 4,000 iterations during training for all the layers in the model.

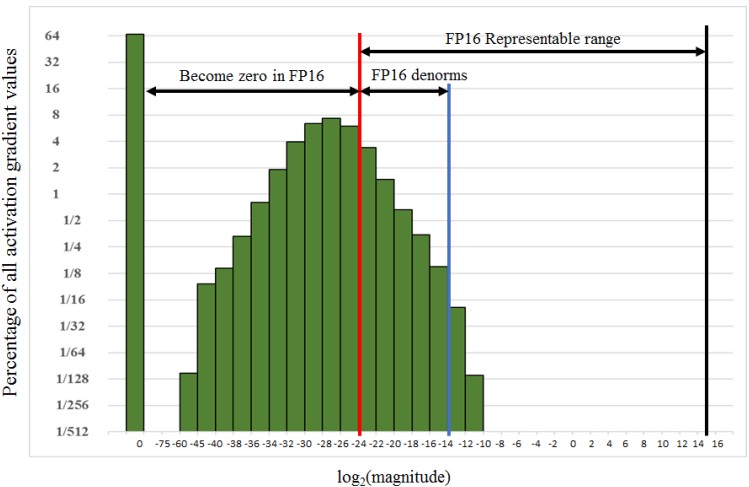

Figure 3: Histogram of activation gradient values during the training of Multibox SSD network. Note that the bins on the x-axis cover varying ranges and there's a separate bin for zeros. For example, 2% of the values are in the $[2^{-34}, 2^{-32})$ range, 2% of values are in the $[2^{-24}, 2^{-23})$ range, and 67% of values are zero.

gradient values below $2^{-27}$ in magnitude were irrelevant to the training of this model, but values in the $[2^{-27}, 2^{-24})$ range were important to preserve.

One efficient way to shift the gradient values into FP16-representable range is to scale the loss value computed in the forward pass, prior to starting back-propagation. By chain rule back-propagation ensures that all the gradient values are scaled by the same amount. This requires no extra operations during back-propagation and keeps the relevant gradient values from becoming zeros. Weight gradients must be unscaled before weight update to maintain the update magnitudes as in FP32 training. It is simplest to perform this unscaling right after the backward pass but before gradient clipping or any other gradient-related computations, ensuring that no hyper-parameters (such as gradient clipping threshold, weight decay, etc.) have to be adjusted.

There are several options to choose the loss scaling factor. The simplest one is to pick a constant scaling factor. We trained a variety of networks with scaling factors ranging from 8 to 32K (many networks did not require a scaling factor). A constant scaling factor can be chosen empir-

ically or, if gradient statistics are available, directly by choosing a factor so that its product with the maximum absolute gradient value is below 65,504 (the maximum value representable in FP16). There is no downside to choosing a large scaling factor as long as it does not cause overflow during back-propagation - overflows will result in infinities and NaNs in the weight gradients which will irreversibly damage the weights after an update. Note that overflows can be efficiently detected by inspecting the computed weight gradients, for example, when weight gradient values are unscaled. One option is to skip the weight update when an overflow is detected and simply move on to the next iteration.

### 3.3 ARITHMETIC PRECISION

By and large neural network arithmetic falls into three categories: vector dot-products, reductions, and point-wise operations. These categories benefit from different treatment when it comes to reduced precision arithmetic. To maintain model accuracy, we found that some networks require that FP16 vector dot-product accumulates the partial products into an FP32 value, which is converted to FP16 before writing to memory. Without this accumulation in FP32, some FP16 models did not match the accuracy of the baseline models. Whereas previous GPUs supported only FP16 multiply-add operation, NVIDIA Volta GPUs introduce Tensor Cores that multiply FP16 input matrices and accumulate products into either FP16 or FP32 outputs (NVIDIA, 2017).

Large reductions (sums across elements of a vector) should be carried out in FP32. Such reductions mostly come up in batch-normalization layers when accumulating statistics and softmax layers. Both of the layer types in our implementations still read and write FP16 tensors from memory, performing the arithmetic in FP32. This did not slow down the training process since these layers are memory-bandwidth limited and not sensitive to arithmetic speed.

Point-wise operations, such as non-linearities and element-wise matrix products, are memory-bandwidth limited. Since arithmetic precision does not impact the speed of these operations, either FP16 or FP32 math can be used.

## 4 RESULTS

We have run experiments for a variety of deep learning tasks covering a wide range of deep learning models. We conducted the following experiments for each application:

- **Baseline (FP32)** : Single-precision storage is used for activations, weights and gradients. All arithmetic is also in FP32.
- **Mixed Precision (MP)**: FP16 is used for storage and arithmetic. Weights, activations and gradients are stored using in FP16, an FP32 master copy of weights is used for updates. Loss-scaling is used for some applications. Experiments with FP16 arithmetic used Tensor Core operations with accumulation into FP32 for convolutions, fully-connected layers, and matrix multiplies in recurrent layers.

The Baseline experiments were conducted on NVIDIA's Maxwell or Pascal GPU. Mixed Precision experiments were conducted on Volta V100 that accumulates FP16 products into FP32. The mixed precision speech recognition experiments (Section 4.3) were conducted using Maxwell GPUs using FP16 storage only. This setup allows us to emulate the TensorCore operations on non-Volta hardware. A number of networks were trained in this mode to confirm that resulting model accuracies are equivalent to MP training run on Volta V100 GPUs. This is intuitive since MP arithmetic was accumulating FP16 products into FP32 before converting the result to FP16 on a memory write.

### 4.1 CNNS FOR ILSVRC CLASSIFICATION

We trained several CNNs for ILSVRC classification task (Russakovsky et al., 2015) using mixed precision: Alexnet, VGG-D, GoogLeNet, Inception v2, Inception v3, and pre-activation Resnet-50. In all of these cases we were able to match the top-1 accuracy of baseline FP32 training session using identical hyper-parameters. Networks were trained using Caffe (Jia et al., 2014) framework modified to use Volta TensorOps, except for Resnet50 which used PyTorch (Paszke et al., 2017).

Training schedules were used from public repositories, when available (training schedule for VGG-D has not been published). Top-1 accuracy on ILSVRC validation set are shown in Table 1. Baseline (FP32) accuracy in a few cases is different from published results due to single-crop testing and a simpler data augmentation. Our data augmentation in Caffe included random horizontal flipping and random cropping from 256x256 images, Resnet50 training in PyTorch used the full augmentation in the training script from PyTorch vision repository.

Table 1: ILSVRC12 classification top-1 accuracy.

| Model | Baseline | Mixed Precision | Reference |
|---|---|---|---|
| AlexNet | 56.77% | 56.93% | (Krizhevsky et al., 2012) |
| VGG-D | 65.40% | 65.43% | (Simonyan and Zisserman, 2014) |
| GoogLeNet (Inception v1) | 68.33% | 68.43% | (Szegedy et al., 2015) |
| Inception v2 | 70.03% | 70.02% | (Ioffe and Szegedy, 2015) |
| Inception v3 | 73.85% | 74.13% | (Szegedy et al., 2016) |
| Resnet50 | 75.92% | 76.04% | (He et al., 2016b) |

Loss-scaling technique was not required for successful mixed precision training of these networks. While all tensors in the forward and backward passes were in FP16, a master copy of weights was updated in FP32 as outlined in Section 3.1.

## 4.2 DETECTION CNNS

Object detection is a regression task, where bounding box coordinate values are predicted by the network (compared to classification, where the predicted values are passed through a softmax layer to convert them to probabilities). Object detectors also have a classification component, where probabilities for an object type are predicted for each bounding box. We trained two popular detection approaches: Faster-RCNN (Ren et al., 2015) and Multibox-SSD (Liu et al., 2015a). Both detectors used VGG-16 network as the backbone. Models and training scripts were from public repositories (Girshick; Liu). Mean average precision (mAP) was computed on Pascal VOC 2007 test set. Faster-RCNN was trained on VOC 2007 training set, whereas SSD was trained on a union of VOC 2007 and 2012 data, which is the reason behind baseline mAP difference in Table 2.

Table 2: Detection network average mean precision.

| Model | Baseline | MP without loss-scale | MP with loss-scale |
|---|---|---|---|
| Faster R-CNN | 69.1% | 68.6% | 69.7% |
| Multibox SSD | 76.9% | diverges | 77.1% |

As can be seen in table 2, SSD detector failed to train in FP16 without loss-scaling. By losing small gradient values to zeros, as described in Section 3.2, poor weights are learned and training diverges. As described in Section 3.2, loss-scaling factor of 8 recovers the relevant gradient values and mixed-precision training matches FP32 mAP.

## 4.3 SPEECH RECOGNITION

We explore mixed precision training for speech data using the DeepSpeech 2 model for both English and Mandarin datasets. The model used for training on the English dataset consists of two 2D convolution layers, three recurrent layers with GRU cells, 1 row convolution layer and Connectionist temporal classification (CTC) cost layer (Graves et al., 2006). It has approximately 115 million parameters. This model is trained on our internal dataset consisting of 6000 hours of English speech. The Mandarin model has a similar architecture with a total of 215 million parameters. The Mandarin model was trained on 2600 hours of our internal training set. For these models, we run the Baseline and Pseudo FP16 experiments. All the models were trained for 20 epochs using Nesterov Stochastic Gradient Descent (SGD). All hyper-parameters such as learning rate, annealing schedule and momentum were the same for baseline and pseudo FP16 experiments. Table 3 shows the results of these experiments on independent test sets.

Table 3: Character Error Rate (CER) using mixed precision training for speech recognition. English results are reported on the WSJ '92 test set. Mandarin results are reported on our internal test set.

| Model/Dataset | Baseline | Mixed Precision |
|---|---|---|
| English | 2.20 | 1.99 |
| Mandarin | 15.82 | 15.01 |

Similar to classification and detection networks, mixed precision training works well for recurrent neural networks trained on large scale speech datasets. These speech models are the largest models trained using this technique. Also, the number of time-steps involved in training a speech model are unusually large compared to other applications using recurrent layers. As shown in table 3, Pseudo FP16 results are roughly 5 to 10% better than the baseline. This suggests that the half-precision storage format may act as a regularizer during training.

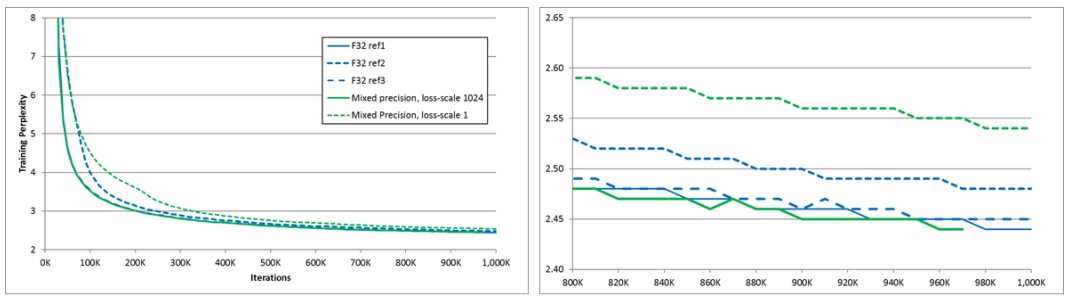

Figure 4: English to French translation network training perplexity, 3x1024 LSTM model with attention. Ref1, ref2 and ref3 represent three different FP32 training runs.

## 4.4 MACHINE TRANSLATION

For language translation we trained several variants of the model in TensorFlow tutorial for English to French translation (Google). The model used word-vocabularies, 100K and 40K entries for English and French, respectively. The networks we trained had 3 or 5 layers in the encoder and decoder, each. In both cases a layer consisted of 1024 LSTM cells. SGD optimizer was used to train on WMT15 dataset. There was a noticeable variation in accuracy of different training sessions with the same settings. For example, see the three FP32 curves in Figure 4, which shows the 3-layer model. Mixed-precision with loss-scaling matched the FP32 results, while no loss-scaling resulted in a slight degradation in the results. The 5-layer model exhibited the same training behavior.

## 4.5 LANGUAGE MODELING

We trained English language model, designated as bigLSTM (Jozefowicz et al., 2016), on the 1 billion word dataset. The model consists of two layers of 8192 LSTM cells with projection to a 1024-dimensional embedding. This model was trained for 50 epochs using the Adagrad optimizer. The the vocabulary size is 793K words. During training, we use a sampled softmax layer with 8K negative samples. Batch size aggregated over 4 GPUs is 1024. To match FP32 perplexity training this network with FP16 requires loss-scaling, as shown in Figure 5. Without loss scaling the training perplexity curve for FP16 training diverges, compared with the FP32 training, after 300K iterations. Scaling factor of 128 recovers all the relevant gradient values and the accuracy of FP16 training matches the baseline run.

## 4.6 DCGAN RESULTS

Generative Adversarial Networks (GANs) combine regression and discrimination tasks during training. For image tasks, the generator network regresses pixel colors. In our case, the generator predicts three channels of 8-bit color values each. The network was trained to generate 128x128 pixel images of faces, using DCGAN methodology (Radford et al., 2015) and CelebFaces dataset (Liu et al.,

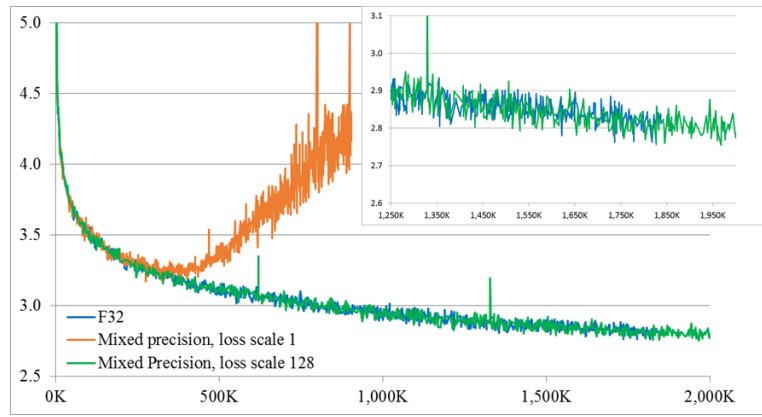

Figure 5: bigLSTM training perplexity

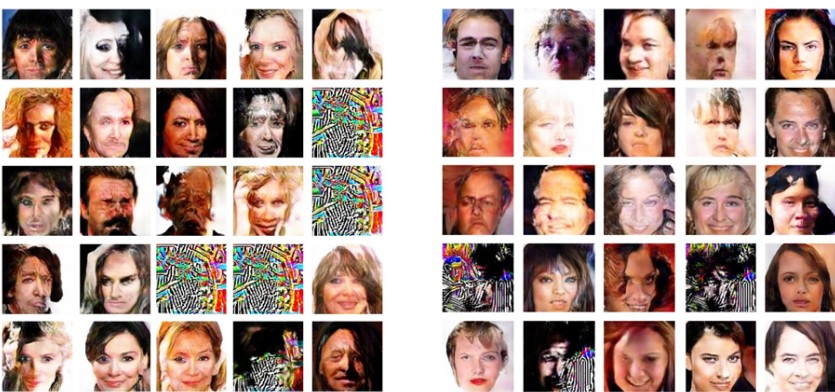

Figure 6: An uncurated set of face images generated by DCGAN. FP32 training (left) and mixed-precision training (right).

2015b). The generator had 7 layers of fractionally-strided convolutions, 6 with leaky ReLU activations, 1 with $tanh$. The discriminator had 6 convolutions, and 2 fully-connected layers. All used leaky ReLU activations except for the last layer, which used sigmoid. Batch normalization was applied to all layers except the last fully-connected layer of the discriminator. Adam optimizer was used to train for 100K iterations. An set of output images in Figure 6. Note that we show a randomly selected set of output images, whereas GAN publications typically show a curated set of outputs by excluding poor examples. Unlike other networks covered in this paper, GANs do not have a widely-accepted quantification of their result quality. Qualitatively the outputs of FP32 and mixed-precision training appear comparable. This network did not require loss-scaling to match FP32 results.

## 5  CONCLUSIONS AND FUTURE WORK

Mixed precision training is an important technique that allows us to reduce the memory consumption as well as time spent in memory and arithmetic operations of deep neural networks. We have demonstrated that many different deep learning models can be trained using this technique with no loss in accuracy without any hyper-parameter tuning. For certain models with a large number of small gradient values, we introduce the gradient scaling method to help them converge to the same accuracy as FP32 baseline models.

DNN operations benchmarked with DeepBench[1] on Volta GPU see 2-6x speedups compared to FP32 implementations if they are limited by memory or arithmetic bandwidth. Speedups are lower when operations are latency-limited. Full network training and inference speedups depend on library

---

[1]https://github.com/baidu-research/DeepBench

and framework optimizations for mixed precision and are a focus of future work (experiments in this paper were carried out with early versions of both libraries and frameworks).

We would also like to extend this work to include generative models like text-to-speech systems and deep reinforcement learning applications. Furthermore, automating loss-scaling factor selection would further simplify training with mixed precision. Loss-scaling factor could be dynamically increased or decreased by inspecting the weight gradients for overflow, skipping weight updates when an overflow is detected.

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
