# OpenReview forum: "Mixed Precision Training"
_ICLR.cc/2018/Conference — Accept (Poster)_

### Official Review · AnonReviewer1 · 2017-11-26
**Exhaustive experiments validate simple techniques**

**Rating:** 8
**Confidence:** 4

**Review:**

The paper considers the problem of training neural networks in mixed precision (MP), using both 16-bit floating point (FP16) and 32-bit floating point (FP32). The paper proposes three techniques for training networks in mixed precision: first, keep a master copy of network parameters in FP32; second, use loss scaling to ensure that gradients are representable using the limited range of FP16; third, compute dot products and reductions with FP32 accumulation.

Using these techniques allows the authors to match the results of traditional FP32 training on a wide variety of tasks without modifying any training hyperparameters. The authors show results on ImageNet classification (with AlexNet, VGG, GoogLeNet, Inception-v1, Inception-v3, and ResNet-50), VOC object detection (with Faster R-CNN and Multibox SSD), speech recognition in English and Mandarin (with CNN+GRU), English to French machine translation (with multilayer LSTMs), language modeling on the 1 Billion Words dataset (with a bigLSTM), and generative adversarial networks on CelebFaces (with DCGAN).

Pros:
- Three simple techniques to use for mixed-precision training
- Matches performance of traditional FP32 training without modifying any hyperparameters
- Very extensive experiments on a wide variety of tasks

Cons:
- Experiments do not validate the necessity of FP32 accumulation
- No comparison of training time speedup from mixed precision

With new hardware (such as NVIDIA’s Volta architecture) providing large computational speedups for MP computation, I expect that MP training will become standard practice in deep learning in the near future. Naively porting FP32 training recipes can fail due to the reduced numeric range of FP16 arithmetic; however by adopting the techniques of this paper, practitioners will be able to migrate their existing FP32 training pipelines to MP without modifying any hyperparameters. I expect these techniques to be hugely impactful as more people begin migrating to new MP hardware.

The experiments in this paper are very exhaustive, covering nearly every major application of deep learning. Matching state-of-the-art results on so many tasks increases my confidence that I will be able to apply these techniques to my own tasks and architectures to achieve stable MP training.

My first concern with the paper is that there are no experiments to demonstrate the necessity of FP32 accumulation. With an FP32 master copy of the weights and loss scaling, can all arithmetic be performed solely in FP16, or are there some tasks where training will still diverge?

My second concern is that there is no comparison of training-time speedup using MP. The main reason that MP is interesting is because new hardware promises to accelerate it. If people are willing to endure the extra engineering overhead of implementing the techniques from this paper, what kind of practical speedups can they expect to see from their workloads? NVIDIA’s marketing material claims that the Tensor Cores in the V100 offer an 8x speedup over its general-purpose CUDA cores (https://www.nvidia.com/en-us/data-center/tesla-v100/). Since in this paper some operations are performed in FP32 (weight updates, batch normalization) and other operations are bound by memory and not compute bandwidth, what kinds of speedups do you see in practice when moving from FP32 to MP on V100?

My other concerns are minor. Mandarin speech recognition results are reported on “our internal test set”. Is there any previously published work on this dataset, or any publicly available test set for this task?

The notation around the Inception architectures should be clarified. According to [3] and [4], “Inception-v1” and “GoogLeNet” both refer to the architecture used in [1]. The architecture used in [2] is referred to as “BN-Inception” by [3] and “Inception-v2” by [4]. “Inception-v3” is the architecture from [3], which is not currently cited. To improve clarity in Table 1, I suggest renaming “GoogLeNet” to “Inception-v1”, changing “Inception-v1” to “Inception-v2”, and adding explicit citations to all rows of the table.

In Section 4.3 the authors note that “half-precision storage format may act as a regularizer during training”. Though the effect is most obvious from the speech recognition experiments in Section 4.3, MP also achieves slightly higher performance than baseline for all ImageNet models but Inception-v1 and for both object detection models; these results add support to the idea of FP16 as a regularizer.

Minor typos:
Section 3.3, Paragraph 3: “either FP16 or FP16 math” -> “either FP16 or FP32 math”
Section 4.1, Paragraph 4: “ pre-ativation” -> “pre-activation”

Overall this is a strong paper, and I believe that it will be impactful as MP hardware becomes more widely used.


References

[1] Szegedy et al, “Going Deeper with Convolutions”, CVPR 2015
[2] Ioffe and Szegedy, “Batch Normalization: Accelerating Deep Network Training by Reducing Internal Covariate Shift”, ICML 2015
[3] Szegedy et al, “Rethinking the Inception Architecture for Computer Vision”, CVPR 2016
[4] Szegedy et al, “Inception-v4, Inception-ResNet and the Impact of Residual Connections on Learning”, ICLR 2016 Workshop

---

> ### Author Response · Authors · 2018-01-05
> **Review Response**
>
> We thank the reviewer for the feedback.  As newer hardware such as Nvidia’s V100 and TitanV become more widely available, we should be able to see a speedup in training time. Performance results for GEMM, RNN, and CNN layers are available at DeepBench. Depending on the layer size and batch size, MP hardware can achieve 2~6x speedup for a given layer, we will add this information to the paper. We are working on measuring the improvement in end-to-end model training using MP hardware and more optimized libraries and frameworks - the studies in this paper used either older hardware, or Volta GPUs but very early libraries and frameworks with MP support.  These measurements are targeted for a subsequent publication as they couldn’t make the ICLR deadline.
>
> When it comes to the need for FP32 accumulation, while some networks did not need it others lost a few percentage points of accuracy when accumulating in fp16.  We will add this mention to the paper, but to maximize the success of initial training in MP we recommend employing all three proposed techniques.
>
> Thank you for pointing out typos, we will address them in this paper.

---

### Official Review · AnonReviewer3 · 2017-11-27
**Mixed Precision Training**

**Rating:** 5
**Confidence:** 3

**Review:**

The paper provides methods for training deep networks using half-precision floating point numbers without losing model accuracy or changing the model hyper-parameters. The main ideas are to use a master copy of weights when updating the weights, scaling the loss before back-prop and using full precision variables to store products. Experiments are performed on a large number of state-of-art deep networks, tasks and datasets which show that the proposed mixed precision training does provide the same accuracy at half the memory.

Positives
- The experimental evaluation is fairly exhaustive on a large number of deep networks, tasks and datasets and the proposed training preserves the accuracy of all the tested networks at half the memory cost.

Negatives
- The overall technical contribution is fairly small and are ideas that are regularly implemented when optimizing systems.
- The overall advantage is only a 2x reduction in memory which can be gained by using smaller batches at the cost of extra compute.

---

> ### Public Comment · ~Stephen_Merity1 · 2017-12-08
> **Overall advantage is both memory and speed**
>
> You note in negatives that "the overall advantage is only a 2x reduction in memory". The paper notes (though only in the introduction section) that "Performance (speed) ... is limited by one of three factors: arithmetic bandwidth, memory bandwidth, or latency", with reduced precision helping two. Specifically, FP16 improves memory bandwidth by only requiring half the data to be shuffled about and that on modern GPUs the FP16 throughput can be 2 to 8 times faster than FP32. Hence, the potential benefit is actually far more than just reducing memory, though the methods and techniques noted in the paper are required in order to have models that can sanely train using FP16.

---

> > ### Author Response · Authors · 2018-01-05
> > **Thanks**
> >
> > Hello Stephen,
> >
> > Thanks for pointing out that the advantage of this technique is not limited to reduction in memory only. We will add some more statements in the paper highlighting the potential speedup with hardware that supports mixed precision training.

---

> ### Author Response · Authors · 2018-01-05
> **Review Response**
>
> Thank you for your review and valuable feedback.  We are working on obtaining speedup numbers for mixed precision training with libraries and training frameworks that have been more extensively optimized for mixed precision (experiments in this study that were run on Volta GPUs used libraries and frameworks that had preliminary optimization for mixed precision).
>
> Initial performance numbers are available in DeepBench which indicate a 2~6x speedup for an operation depending on layer size and batch size, as long as the layer is not limited by latency (as stated in the paper, mixed-precision improves performance for 2 out of 3 potential performance limiters - memory or arithmetic throughput, with latency being the third one).  For layers limited by memory bandwidth, as you point out, upper bound on speedup is 2x. The upper bound on speedups on Volta GPUs is 8x, if the operation is limited by floating point arithmetic. Full network speedups will be somewhat lower, depending on how many layers are limited by memory bandwidth or latency.

---

### Official Review · AnonReviewer2 · 2017-11-29
**Computation improvements for training neural nets**

**Rating:** 7
**Confidence:** 4

**Review:**

The paper presents three techniques to train and test neural networks using half precision format (FP16) while not losing accuracy. This allows to train and compute networks faster, and potentially create larger models that use less computation and energy.

The proposed techniques are rigorously evaluated in several tasks, including CNNs for classification and object detection, RNNs for machine translation, language generation and speech recognition, and generative adversarial networks. The paper consistently shows that the accuracy of training and validation matches the baseline using single precision (FP32), which is the common practice.

The paper is missing results comparing training and testing speeds in all these models, to illustrate the benefits of using the proposed techniques. It would be very valuable to add the baseline wall-time to the tables, together with the obtained wall-time for training and testing using the proposed techniques.

---

> ### Author Response · Authors · 2018-01-05
> **Review Response**
>
> Thank you for the review and comments.  The focus of the studies in the paper, as you point out, was to describe and validate the procedure for training with mixed precision without losing accuracy.  Experiments were run with libraries and frameworks that had preliminary support for mixed precision.  As shown in DeepBench (https://github.com/baidu-research/DeepBench), depending on the layer size and batch size, MP hardware can achieve 2~6x speedup for a layer that’s not latency-limited. We will add this mention and pointer to DeepBench results to the paper.  Measuring end to end speedups with more optimized frameworks is the focus for future work.

---

### Author Response · Authors · 2018-01-05
**New Revision of the paper**

We've added a new revision of the paper that addresses the following points:

- Corrected the typos pointed out by reviewer #1
- Added a reference to Inception v3, modified the classification CNN table to include references and both names for googlenet.  - Added improved accuracy numbers for ResNet-50
- Added a paragraph to the conclusion on operation speedups, etc.


We thank the reviewers for their helpful comments and feedback.

---

### Decision · Program_Chairs · 2018-01-29
**ICLR 2018 Conference Acceptance Decision**

**Decision:**

Accept (Poster)

**Comment:**

meta score: 8

The paper explores mixing 16- and 32-bit floating point arithmetic for NN training with CNN and LSTM experiments on a variety of tasks

Pros:
 - addresses an important practical problem
 - very wide range of experimentation, reported in depth

Cons:
 - one might say the novelty was minor, but the novelty comes from the extensive analysis and experiments